

Open Access 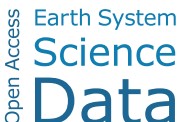 Earth System
Science
Data

# The International Satellite Cloud Climatology Project H-Series climate data record product

**Alisa H. Young**[1], **Kenneth R. Knapp**[2], **Anand Inamdar**[3], **William Hankins**[2,4], **and William B. Rossow**[5]

[1]NOAA's National Centers for Environmental Information, 325 S. Broadway, Boulder, CO 80305, USA
[2]NOAA's National Centers for Environmental Information, 151 Patton Ave, Asheville, NC 28801, USA
[3]Cooperative Institute for Climate and Satellites, North Carolina State University, USA TS1
[4]ERT, Inc., Asheville, NC 28801, USA
[5]NOAA/CREST, City College of the City University of New York, New York, NY 10031, USA

**Correspondence:** Alisa H. Young (alisa.young@noaa.gov)

**Abstract.** This paper describes the new global long-term, International Satellite Cloud Climatology Project (ISCCP) H-series climate data record (CDR). The H-series data contain a suite of level 2 and 3 products for monitoring the distribution and variation of cloud and surface properties to better understand the effects of clouds on climate, the radiation budget, and the global hydrologic cycle. This product is currently available for public use and is derived from both geostationary and polar-orbiting satellite imaging radiometers with common visible and infrared (IR) channels. The H-series data currently span July 1983 to December 2009 with plans for continued production to extend the record to the present with regular updates. The H-series data are the longest combined geostationary and polar orbiter satellite-based CDR of cloud properties. Access to the data is provided in network common data form (netCDF) and archived by NOAA's National Centers for Environmental Information (NCEI) under the satellite Climate Data Record Program (https://doi.org/10.7289/V5QZ281S TS2). The basic characteristics, history, and evolution of the dataset are presented herein with particular emphasis on and discussion of product changes between the H-series and the widely used predecessor D-series product which also spans from July 1983 through December 2009. Key refinements included in the ISCCP H-series CDR are based on improved quality control measures, modified ancillary inputs, higher spatial resolution input and output products, calibration refinements, and updated documentation and metadata to bring the H-series product into compliance with existing standards for climate data records.

## 1 Introduction

The International Satellite and Cloud Climatology Project (ISCCP) was established in 1982. Its intent was to produce a global, reduced-resolution, calibrated infrared and visible radiance dataset with basic information on surface and atmospheric radiative properties and to derive global cloud characteristics from satellite data (Schiffer and Rossow, 1983). Today, ISCCP is the longest-running international satellite-based global environmental data project. It delivers a record spanning over 25 years of global cloud and surface radiative properties obtained from radiance images from geostationary and polar-orbiting satellites. As a mark of the dataset's

value, it has been cited in more than 15 000 articles, with Rossow and Schiffer (1999) receiving over 1800 citations (Fig. 1) and continuing. This achievement can be attributed to the precedent set by the World Climate Research Program that established ISCCP and utilized international collaborations to obtain, process, distribute, and archive data from US- and non-US-operated geostationary and polar imaging meteorological satellites. The collection of ISCCP applications and analyses demonstrate that ISCCP has made a significant contribution to advancing climate science and assessment. However, the widely used ISCCP D-series product has not been updated since December 2009. Moreover, several stud-

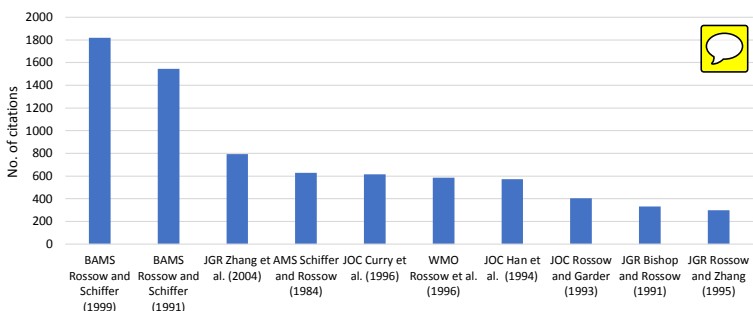

**Figure 1.** TS3 ISCCP ten most cited papers that have contributed to the dataset's more than 15 000 citations. The number of citations given here is based on Google Analytics TS4

ies have evaluated the product to highlight specific opportunities to advance the dataset (Rossow and Ferrier, 2015; Evan et al., 2007; Norris, 2000; Rossow and Schiffer, 1999; Stubenrauch et al., 2013) and take further advantage of its record, spanning over 25 years, to improve its capability to estimate long-term trends in global cloudiness. This detail is relevant considering newer cloud datasets that have shorter records but improved capabilities for cloud detection and retrieval due to technological advancements that include active spaceborne sensors (e.g., Cloud–Aerosol Lidar and Infrared Pathfinder Satellite Observations – CALIPSO – and Cloud-Sat) and cloud datasets that rely on newer passive imagers with higher spectral, spatial, radiometric, and temporal resolutions (Platnick et al., 2003; Hutchison et al., 2005; Stengel et al., 2017).

To build on ISCCP's legacy and further advance the dataset in light of these advancements, in 2004, a large data stewardship effort by the National Climatic Data Center (now known as the National Centers for Environmental Information – NCEI) led to the rescue of ISCCP B1 data with ∼ 10 km and 3-hourly spatial and temporal resolution (Knapp, 2008). This effort set the stage for ISCCP B1U (uniformly formatted B1) data to serve as the new geostationary satellite data input to ISCCP processing. The NASA MEaSUREs (Making Earth Science Data Records for Use in Research Environments) and NOAA climate data record programs have served as resources for implementing product updates that exploit the higher resolution B1U and global area coverage (GAC) AVHRR data and more recent research results. The latter includes results from the Global Energy and Water Cycle Experiment (GEWEX) cloud assessment in which a special version of the ISCCP D-series level 3 monthly product with 1° spatial resolution was compared with 11 other "state-of-the-art" cloud datasets from active and passive remote sensors (Stubenrauch et al., 2012, 2013). Relative geographical and seasonal variations in the cloud properties agree very well (with only a few exceptions, like deserts and snow-covered regions). Discrepancies among the various products for detection and retrieval of cloud properties were mainly due to the use of different spectral domains

and instrument performance. However, some of the results from these and other evaluations (e.g., Evan et al., 2007; Jiménez, et al., 2012) have led to algorithmic changes for production of ISCCP H-series data described herein.

To document these updates, this paper gives a description of the new ISCCP H-series product with specific emphasis on the changes in the algorithm and products in transitioning from the D-series (Rossow and Schiffer, 1999) to the H-series. The more complete version of all the product updates are contained in the Climate-Algorithm Theoretical Basis Document (Rossow, 2017). The sections below provide a description of the newly developed H-series collection, comparison with its predecessor D-series product, details for data access, caveats, and plans for future development under the stewardship of NOAA's NCEI.

## 2 ISCCP H-series processing

Like the ISCCP D-series products, the primary instruments that serve as inputs to the ISCCP H-series analysis are the imaging radiometers on operational weather satellites. These include the Advanced Very High Resolution Radiometer (AVHRR) on the polar-orbiting satellites and a variety of imagers (Rossow, 2017) that fly onboard the geostationary meteorological satellites. ISCCP handles these data using seven data-processing streams. Both the geostationary and polar orbiter (AVHRR GAC) data have been sampled to ∼ 10 km spatial resolution. The ISCCP data-processing streams are labeled by the originating satellites and are provided in Fig. 2, in which the ISCCP general processing for pixel-level cloud detection and retrievals is illustrated. The seven data-processing streams are given by the following.

- GMS: Japanese Geostationary Meteorological Satellite with a subsatellite longitude of ∼ 140° E;

- INS: Indian ocean sector coverage with a subsatellite longitude at ∼ 63° E;

- MET: European and African sector coverage with a subsatellite longitude of ∼ 0°.

Please note the remarks at the end of the manuscript.

**ISCCP data flow overview**

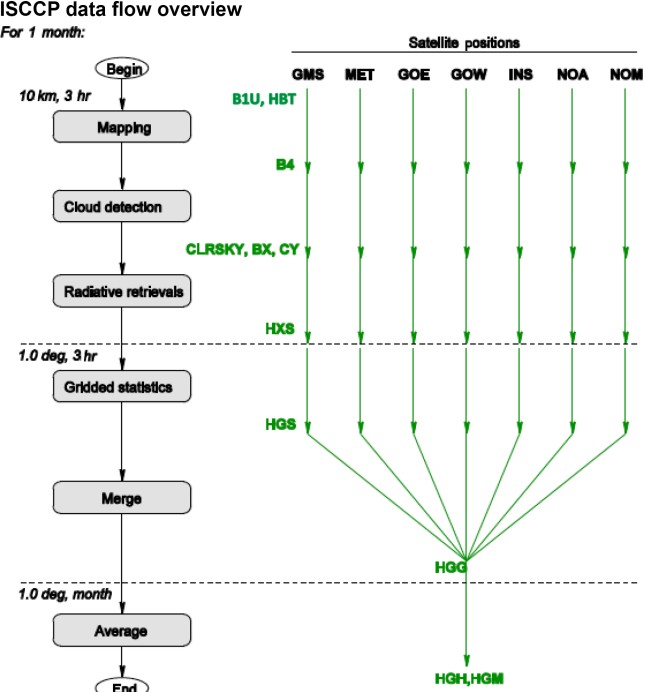

**Figure 2.** Illustration of ISCCP production with satellite processing streams defined for five geostationary data streams (GMS at 140° E, MET at 0°, GOE at 75° W, GOW at 135° W and INS at ~ 63° E) and two polar orbiter streams (NOM and NOA). The left side of the image shows important steps in ISCCP H-series data processing that feed into the various H-series products.

- GOE: Eastern United States and South American coverage with a subsatellite longitude of 75° W;

- GOW: Pacific Ocean and western United States coverage with a subsatellite longitude of 135° W;

5 - NOA: afternoon polar-orbiting satellite stream;

- NOM: morning polar-orbiting satellite stream.

TS6 In the mapping step, data are mapped to a 10 km grid. Geostationary data are preferred between 55° N and 55° S. If more than one geostationary satellite is available, the geo-10 stationary satellite with the larger cosine of the satellite view zenith angle is preferred. The afternoon polar orbiter results are used if no geostationary results are available and, finally, the morning polar orbiter is used if no geostationary or afternoon polar orbiter data are available. Likewise, po-15 lar orbiter data are preferred poleward of 55° N/S but may rely on geostationary results in the absence of PO data. The combination of the geostationary and polar-orbiting satellites allows ISCCP to establish an intercalibration procedure in which radiances from imagers onboard the geostationary 20 satellites are normalized to the low-earth-orbit AVHRR radiances from the afternoon polar orbiter satellite series. In this approach, NOAA-9 acts as the absolute reference through

2009 (Rossow and Ferrier, 2015). As the H-series dataset is processed forward in time, NOAA-18 will serve this function. Although most of the imaging radiometers make measurements of radiation emitted from earth at multiple spec-25 tral wavelengths, the H-series product uses only one visible (VIS ≈ 0.65 ± 0.05–0.20 μm) and infrared (IR ≈ 10.5 ± 0.5–0.75 μm) "window" channel to derive cloud and surface properties. In previous versions of the ISCCP, data products 30 have relied on B3 data with 3-hourly and 30 km temporal and spatial resolution (Rossow and Schiffer, 1985 TS7). However, the primary geostationary input to ISCCP H-series is B1U data, which have 3-hourly and ~ 10 km temporal and spatial resolutions CE1 . ISCCP ancillary products have also 35 undergone modifications following recommendations from Raschke et al. (2006). Table 1 shows the details of D- to H-series ancillary product changes. In general, the updated input and ancillary data products yield a more consistent record for the reprocessing of higher resolution cloud products. 40

## 3 ISCCP H-series cloud detection

The ISCCP H-series cloud detection algorithm and retrievals are generally minor revisions of the D-series algorithm and retrievals that mostly serve to reduce uncertainties. The algorithm is largely described by four steps following the re-45 mapping step shown in Fig. 2. First, tests of the space and time variations of the observed radiances on several scales are used to estimate cloud-free radiances (B4). Results of the space–time tests are used in conjunction with the ancillary products to obtain a global composite of clear-sky radiances 50 for each image pixel location and time (CLRSKY). Second, cloudy conditions are diagnosed when IR- or VIS-observed satellite radiances sufficiently deviate from estimated values using various combinations of VIS and IR thresholds (BX) (Rossow and Garder, 1993a, b; Rossow et al., 1993). From 55 here, the composite clear-sky radiances are revised based on the prior detection threshold results and application of revised threshold tests of each image's pixels against the revised composite clear-sky radiance values using the ancillary products (CY). Then finally, cloud and surface prop-60 erties are retrieved producing the HXS product (cf. CE2 , Rossow and Schiffer, 1991, 1999). These steps summarize the ISCCP processing system subroutines (B4PROD (B4), CLRSKY, BX, and CY) referenced in Fig. 2.

Differences between the D- and H-series cloud detection 65 algorithms include the following modifications: (1) a new radiance space contrast test inside regions of land–water mixtures, (2) updated surface type categories for algorithm tests to improve cloud tests in rough topography, (3) revised daytime cloud detection over snow and ice by eliminating 70 3.7 μm tests since this channel is not available for all AVHRR datasets over the whole period of record and implemented simpler test for reversed VIS radiance contrast situations to improve homogeneity of record, (4) improved summertime

**Table 1.** List of H-series and D-series ancillary data products including in producing ISCCP cloud and surface products; n/a: not applicable TS8.

| | Version | Product | Description | Product reference | Product resolution |
|---|---|---|---|---|---|
| Atmospheric profiles | H | nnHIRS | Neural network analysis of High-resolution Infrared Radiometer Sounder (HIRS) and stratospheric water ozone satellite homogenized data | Shi et al. (2016) | 3-hourly global 1° equivalent equal-area grid |
| | D | TOVS | Atmosphere and surface data including temperature structure, water, and ozone abundances obtained from the TIROS Operational Vertical Sounding (TOVS) Product and supplemented by two climatologies Sounding (TOVS) Product and supplemented by two climatologies. | | Daily 280 km equivalent equal-area grid |
| AEROSOL | H | MACv.1 | Merges surface-based aerosol emission data from AERONET and satellite products from MODIS and MISR, with the median results from an ensemble of emission-transport models. | Kinne et al. (2013) | Monthly 1° equivalent equal-area grid |
| | D | n/a | | | |
| OZONE abundance | H | TOMS, TOVS, SMOBA, OMI | Daily variations of the global distribution of total column ozone abundance from a combination of satellite-based instruments. Data is reported at 16 vertical levels. | Stolarski et al. (1981 TS9), Kroon et al. (2011), Chesters and Neuendorffer (1991) TS10, Neuendorffer et al. (1996), Yan et al. (2006) | Daily 1° equivalent equal-area grid |
| | D | TOVS | The main dataset used to produce a daily, global description of the ozone, temperature, and humidity distributions is that obtained from the analysis of data from the TIROS Operational Vertical Sounder (TOVS) System. Data are reported at 10 vertical levels. | ISCCP Science Team https://doi.org/10.5067/ISCCP/TOVS_NAT | Daily 280 km equivalent equal-area grid |
| SNOW/ICE cover fraction | H | | Northern Hemisphere EASE-Grid Weekly Snow Cover and Sea Ice Extent (Version3), NOAA NSIDC IMS Daily Northern Hemisphere Snow and Ice Analysis, OSI-SAF Global Sea Ice Concentration Reprocessing Dataset, GLIMS permanent glacier cover product. | Brown and Robinson (2011), Armstrong and Brodzik (2005) TS11 | Daily 0.25° equivalent equal-area grid |
| | D | | Averages of snow and sea ice fractional coverage deduced from ship or shore station reports and satellite visible, infrared, and microwave imagery data. | ISCCP Science Team (1999) TS12 | 112 km equal area grid, 5-day, global |
| TOPO | H | | USGS Earth Resources Observation and Science (EROS) GTOPO30 product reconciled with the USGS Global Land 1 km AVHRR Project land–water mask. The data were modified to produce a reconciled product for use in ISCCP. | These data are available from the US Geological Survey | Fixed 0.1° equivalent equal-area grid |
| | D | | Same as H-series* TS13 | | |
| SURFACETYPE | H | | MODIS International Geosphere–Biosphere Programme (IGBP) surface type classification. | Loveland et al. (2009 TS14) | Fixed 0.10° equivalent equal-area grid |
| | D | | Global Vegetation Types, 1971–1982; A global digital database of vegetation was compiled at 1° latitude by 1° longitude resolution, drawing on approximately 100 published sources. | Matthews (1983) https://doi.org/10.3334/ORNLDAAC/419 | Fixed 1.0° |

polar cloud detection by reducing VIS thresholds over snow and ice, and (5) improved wintertime polar cloud detection by changing marginally cloudy to clear and marginally clear to cloudy. Otherwise, the current H version (v01r00) of the ISCCP cloud detection algorithm is the same as the D version which is a modification of the C version. Hence, all publications regarding the first two versions of ISCCP products

are also relevant to the H-series algorithm. Likewise, the differences in the D- and H-series surface and cloud retrievals are generally due to small changes in the assumptions in the radiative transfer calculations on which they are based. The most notable changes are listed in the next section.

**Table 2.** High-level summary of differences between ISCCP D-series and H-series products and their impacts. Other details on differences are provided in the C-ATBD; n/a: not applicable TS15.

| | D-series | H-series | Impacts |
|---|---|---|---|
| Input resolution | 30 km, B3 | 10 km, B1U | Higher spatial resolution output products |
| Cloud algorithm | Combination of IR and VIS, and NIR channels in polar regions | Only IR and VIS for entire period of record | Reduces low-level cloud sensitivity over snow and ice in polar regions |
| Metadata | n/a | CF 1.6 | Improved documentation for product reproducibility |
| Improved/added ancillary data | | Example: Macv1 Aerosols Product | Increases cloud fraction due to volcanic eruptions and reduce cloud optical thicknesses in areas with larger aerosol abundances |
| Period of record | 07/1983–12/2009 with no additional production planned to extend POR | 07/1983–12/2009 with additional production planned to extend POR | Restores research application of ISCCP data post-2009 |
| Format | Binary | netCDF for all products except HXS | Supported by netCDF software applications and tools. |
| Available products | DX ($\sim$ 30 km, 3 hourly) n/a DS D1 (2.5°, 3 hourly) D2 (2.5°, 3-hourly monthly avg.) D3 (2.5°, monthly avg.) | HXS ($\sim$ 10 km, 3 hourly) HXG (0.1°, 3 hourly) HGS (1.0°, 3 hourly) HGG (1.0°, 3 hourly) HGH (1.0°, 3-hourly monthly avg.) HGM (1.0°, monthly avg.) | |

## 4 ISCCP H-series products

### 4.1 H-series products

Table 2 provides a summary of the differences between the ISCCP D- and H-series products. The ISCCP D-series algorithm relied on ISCCP Stage B3 data with spatial and temporal resolutions of 30 km and 3 h for geostationary satellites. Thus, the highest resolution D-series data produced the 30 km 3-hourly product for individual satellites known as DX. Downstream level 3 products included D1 (global and 3 hourly) and D2 (monthly mean) products on an equal area grid with a spatial interval of 280 km (2.5° equivalent). In comparison, the ISCCP H-series products rely on $\sim$ 10 km and 3-hourly B1U data and polar orbiter data sampled to $\sim$ 10 km intervals. The level 2 products are HXS and HXG and level 3 products are HGS, HGG, HGH, and HGM. The products have the following descriptions:

– HXS (H-series pixel level by satellite) provides pixel-level results of cloud and surface properties retrieved or used in the retrieval for each individual satellite image in nearly the original projection for geostationary satellites and for groupings of orbit swaths for polar orbiter data in six midlatitude (ascending and descending swaths in 120° longitude sectors) and two polar sectors.

– HXG (H-series pixel-level global) is a global merger of the information from HXS common to all satellites and is mapped and provided every 3 h on a 0.10° equal angle grid ($\sim$ 240 files per month).

– HGS (H-series gridded by satellite) reduces the HXS Product to the 1° equal-angle grid with additional statistical and cloud type information and combines these results with the information from the ancillary data products prior to the global merger.

– HGG (H-series gridded global) is the global merger of the HGS products from all available satellites (e.g., all HGS files), in which overlapping coverage is resolved in favor of the satellite with the best viewing geometry, with a preference for geostationary results at lower latitudes and polar orbiter results in the polar regions. The time interval is 3 h and the map grid is 1° equal-area grid. The HGG product is the H-series analogue to the D1 product and should be regarded as the main ISCCP Cloud Product.

– HGH (high-resolution global hourly) is the monthly 1° equal-area gridded average of the HGG product at each of the eight 3-hourly times of day (00Z, 03Z, 06Z, etc.) used in the ISCCP algorithm.

- HGM (high-resolution global monthly) is the average of the eight HGH products for each month.

All H-series products, except HXS, are formatted in netCDF-4. Other differences in the D- and H-series products include (1) revisions in the COUNTS CE3 -to-physical conversion tables to remove special values for underflow and overflow; (2) increased uncertainty estimate information; and (3) missing observations are filled in the global, 3-hourly product (HGG) instead of the monthly product (the HXG product is also filled). A subset of the HGG, HGH, and HGM products are also available in a CF CE4 -compliant equal angle format known as ISCCP Basic, which has fewer variables and a total volume of 305 GB. Other changes between the D-series and H-series products include the following. TS16

- Radiance calibrations from D version to H version:

    1. anchor for VIS calibration extended to combine results for NOAA-9 (through 2009) and NOAA-18 (post-2009), spanning the whole record;

    2. overall IR calibration adjusted for small gain error in AVHRR calibrations compared to MODIS for all AVHRRs on NOAA-15 and onward (Cao and Heidinger, 2002).

    3. geostationary normalization procedure changed to use all of the radiance data directly instead of a small number of special samples – manual procedures eliminated (similar to that used by Inamdar and Knapp, 2015) and corrected the AVHRR KLM calibration error after 2001 (Evan et al., 2007).

- VIS and IR Radiance Models from D version to H version:

    1. replaced ocean VIS reflectance model with more accurate version that includes a better glint treatment.

    2. calculated instrument-specific ozone absorption coefficients;

    3. added water vapor above 300 mb level in atmospheric ancillary data;

    4. added treatment of stratospheric and tropospheric aerosol scattering and absorption;

    5. improved surface temperature retrieval by accounting for variations of surface IR emissivity by surface type;

    6. introduced more explicit atmospheric and cloud vertical structures for cloud retrievals;

    7. changed specified liquid cloud droplet effective radius from 10 μm everywhere to 13 and 15 μm over land and ocean, respectively;

    8. changed cloud-top temperature value separating ice and liquid phase clouds from 260 to 253 K;

    9. updated ice cloud scattering phase function to empirically based model from satellite polarimetry observations and revised specified ice particle effective radius from 30 μm for all clouds to 20 and 34 μm for clouds with TAU < 3.55 and TAU ≥ 3.55, respectively;

    10. corrected placement of thin clouds from just above the tropopause to at the tropopause;

    11. added treatment of cloud-top location when surface temperature inversions are present.

    12. updated solar ephemeris.

### 4.2 Product variables

Beginning with the original C-Series product, ISCCP has delivered an extensive set of product variables. The cloud properties include (but are not limited to) the following:

- cloud amount

- cloud-top temperature, TC (in Kelvins)

- cloud-top pressure, PC (in mb)

- cloud optical thickness, TAU (unitless)

- cloud water path, CWP (in $\mathrm{g\,m^{-2}}$)

- cloud phase

- cloud type.

Surface properties include the following:

- surface temperature, TS (in Kelvins)

- surface reflectance, RS (unitless).

Separate procedures are used to produce these data under daytime versus nighttime conditions (the nighttime procedure is applied day and night). In the H-series basic product introduced in Sect. 4.1 these variables are converted to their physical units. For a more detailed list of all ISCCP variables, please refer to the ISCCP Climate-Algorithm Theoretical Basis Document (Rossow, 2017).

## 5 Basic characterization of the ISSCP H-series monthly cloud amount

Given the higher resolution of the B1U data, the H-series data yield cloud characteristics with finer spatial detail and more robust spatial distribution statistics. The improvements to the product take account of recent research results concerning cloud properties that are assumed in the retrieval and enhances its capabilities to assess cloud characteristics and variability that occur on regional to global scales. Some impacts of the changes are illustrated in Fig. 3, which shows the January 2009 monthly mean ISCCP cloud amount for

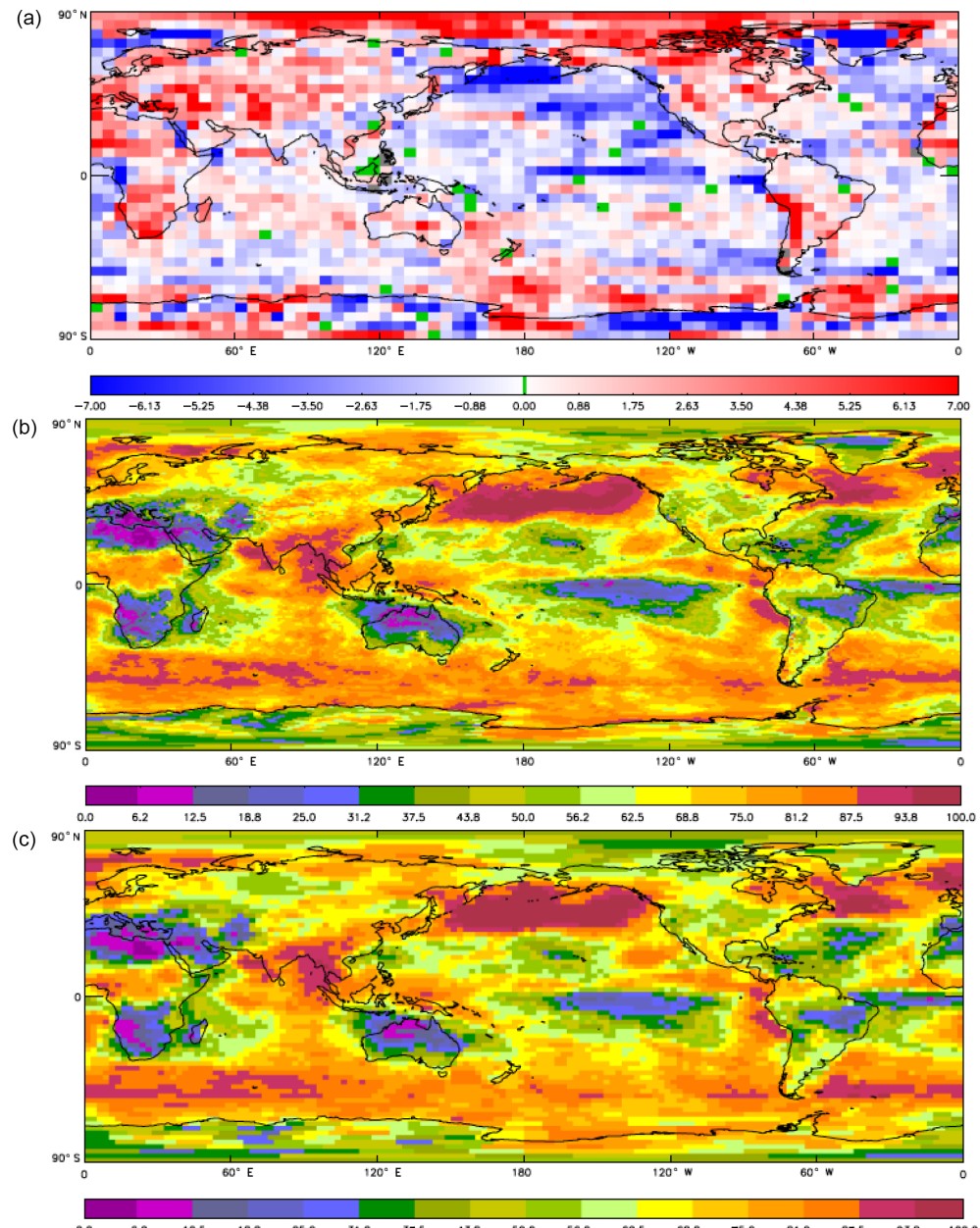

**Figure 3.** January 2009 ISCCP percentage of global cloud amount for **(a)** differences between H- and D-series, **(b)** H-series HGM product at 1° and **(c)** D-series D3 product at 2.5°. As shown, in **(a)** the differences between the products are greatest in the polar and coastal regions where for this case the H-series product has a slightly higher cloud fraction. In general, the H- and D-series distributions of cloud amount have good agreement.

(a) percent (%) differences between H- and D-series, (b) the H-series monthly mean (HGM) product at 1°, and (c) the D-series monthly mean (D3) product at 2.5°. As shown in (a) H- and D-series differences are greater in polar and coastal regions, mostly due to the exclusion of the AVHRR NIR channel (3.7 µm) in the H-series cloud algorithm (see Table 2). Differences are also present due to the higher resolution input (B1U) data, which impacts the assessment of clear or CE5 cloudy scenes (which increases the number of scenes with no cloud cover or total cloud cover), to enhanced efforts to gather and/or limit undesirable radiance images from processing and production via QC, and to changes in the analysis procedure described in Sect. 2. Based upon these differences, the January 2009 HGM product has a slightly lower global mean cloud fraction (cf. 65.46 %, H, and 66.29 %, D). In general, the main cloud properties are very similar on average. However, the grid-scale distributions have more noticeable differences in the ratio of ice- and liquid-phase clouds and

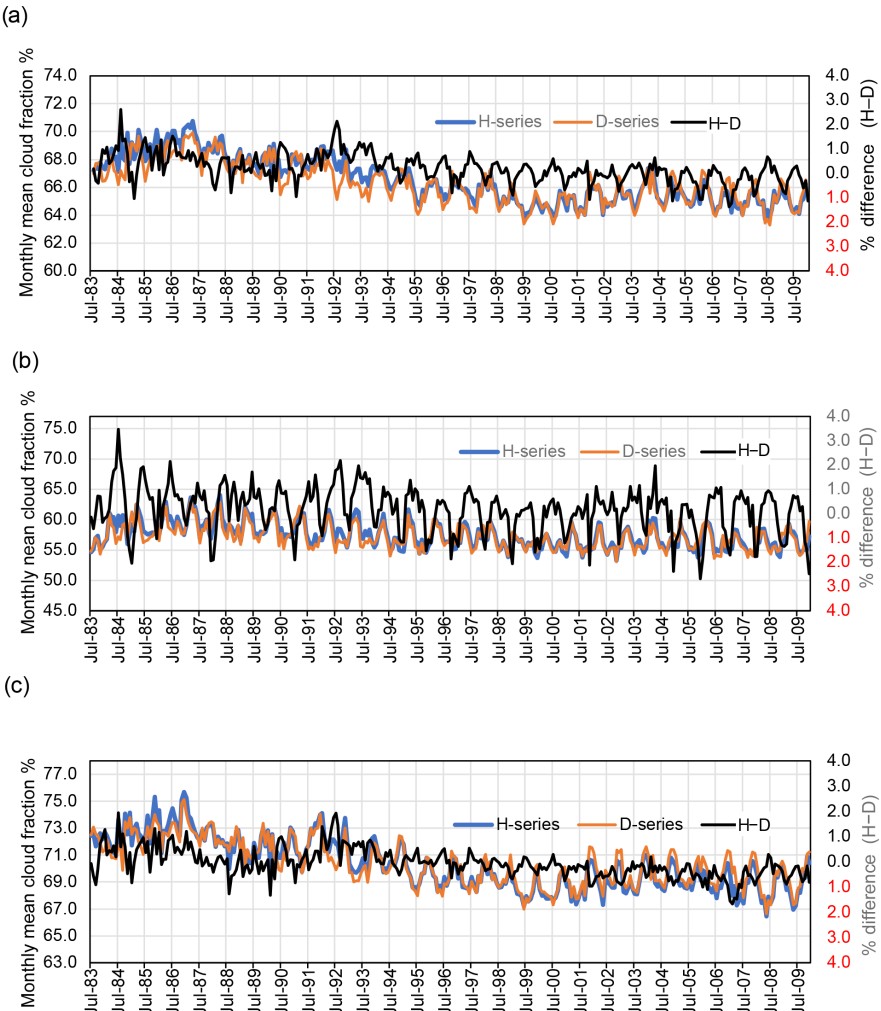

**Figure 4.** Comparison of ISCCP H- (blue) and D-series (orange), and differences between H- and D-series (black) monthly mean cloud fraction ( %) for **(a)** total (land and water), **(b)** land only, and **(c)** water only. For the secondary vertical axes, black numbers represent positive differences and red numbers are negative. Data are for July 1983 through December 2009.

in the optical thicknesses of thicker ice clouds in the polar regions.

In addition to the monthly H- and D-series comparison provided in Fig. 3, which gives users a monthly snapshot of the H- and D-series CF differences (i.e., H–D), Fig. 4 provides the comparison of ISCCP H- and D-series monthly mean cloud fraction (%) for July 1983 through December 2009 for (a) the globe, (b) land, and (c) water. The global mean differences are on average ∼ 0.21 %. This demonstrates that the H-series product generally captures a slightly higher cloud fraction compared to D-series data. However, H- and D-series differences follow a seasonal pattern whereby the average H-series CF for November through April is slightly lower than the D-series product, and during May–October, H-series CF is slightly higher than the D-series product: this difference is due mainly to the impact of the algorithm changes over the polar regions, more sig-

nificantly over Antarctica. As displayed in Fig. 4b and c the monthly mean land cloud fraction for both H- and D-series is generally less than the CF reported for water. The land CF also reflects a higher percentage of the mean differences (0.16 %) compared to water (−0.06 %). Other components of the comparison between H- and D-series data (not shown) reveal that the inclusion of MACv1 for the treatment of stratospheric and tropospheric aerosols reduces the cloud optical thickness in cases of larger aerosol amounts.

## 6 Product caveats

There are some caveats that users should be aware of that primarily involve the absence of some data in the initial release of the product.

The following is a list of issues and caveats users should know.

– General notes:

– Calibration D to H – ISCCP H series calibration follows the method and process of the ISCCP D series. Although a correction is applied for the AVHRR NOAA KLM calibration error, most calibration issues present in ISCCP D are also present in the H-series product. Users may refer to Brest and Rossow (1992) TS17, Desormeaux et al. (1993 TS18), Brest et al. (1997 TS19), Inamdar and Knapp (2015), and Rossow and Ferrier (2015). All these citations, plus many others, are given in the Climate Algorithm Theoretical Basis Document (C-ATBD).

– Spatiotemporal analysis – ISCCP H series cloud algorithm is mostly unchanged. The examination of the geographic distributions of average ISCCP cloud amounts continues to show artifacts in association with large changes in the average value of satellite zenith angle (Rossow and Garder, 1993b).

– Satellite coverage – the ISCCP product is limited by the input geostationary datasets. These have gaps in coverage that are large and small (seen in the geostationary quilt, Knapp et al. (2011). The larger gaps are caused by satellite outages, or gaps in the geostationary ring. The smaller gaps can be up to a week in length and occur more often in the early years.

– Specific issues:

– MET-3 1995 – Many B1U files are missing the visible channel.

– GMS-3 1986 – Many B1U files for February–April are missing the visible channel.

– The afternoon Polar Orbiter data (NOM) has a 2-year gap from 2000 to 2002 for the NOAA-15 to NOAA-17 transition. We have the data and just received status for the AVHRR instrument for this period. This will be resolved in future reprocessing.

– There are occasional cloud-top pressure errors over the Pacific for May 1994 (and possibly other months). This is caused by large-view zenith angles in glint regions.

## 7 Product access, availability, and future development

ISCCP H-series data are currently available for July 1983–December 2009 with plans for updates beginning in December 2017 that will extend the record forward in time to 2015. The record will be operationally maintained with annual updates beginning in 2018. The NOAA Climate Data Record of the ISCCP H-series product, version v01r00, is archived and distributed by NCEI's satellite Climate Data Record Program. The ISCCP H-series products are maintained by and available from NOAA. The full set of IS-CCP CDR products, as well as the ancillary data, are publicly available with points for access given at https://www.ncdc.noaa.gov/isccp/isccp-data-access. The processing code and the Climate Algorithm Theoretical Basis Document (C-ATBD), which more fully outlines ISCCP H-series production, can be accessed from https://www.ncdc.noaa.gov/cdr/atmospheric/cloud-properties-isccp. The ISCCP H-series Basic CDR product can be downloaded via FTP or from the NCEI THREDDS data server (https://doi.org/10.7289/V5QZ281S). Users are also requested to register at https://www.ncdc.noaa.gov/isccp.

The future development of the ISCCP H-series products includes the following:

– setting up the ISCCP system to process the newer geostationary and polar orbiter imagers (e.g., Himawari-8 and GOE S-R) to extend the data record through the present with operational plans for annual updates;

– improvements to satellite calibration particularly to increase the calibration consistency between adjacent geostationary satellites;

– continued efforts for backfilling missing data to develop a more complete record.

## 8 Data availability

. TS20

## 9 Conclusions

ISCCP H-series data are now a component of NOAA's suite of climate data records and will be operationally produced and updated by NOAA NCEI. Research users are encouraged to use the ISCCP products described herein to investigate cloud processes in weather and climate. The ISCCP Basic product is suitable for software applications that allow for ease in viewing and handling netCDF files (i.e., Weather Climate Toolkit, Panoply, ToolsUI, etc.). Future improvements and versions will be driven by user requirements.

**Competing interests.** The authors declare that they have no conflict of interest. TS21

**Acknowledgements.** This project received funding from the NOAA/NCEI Climate Data Record Program and NASA MEaSUREs Program Support. The authors wish to thank TS22 D. Wunder, V. Toner, C. Hutchins, J. Buddai, and the entire ISCCP Integrated Product Team (IPT) at NCEI for supporting the Research to Operations (R2O) process. Special appreciation goes to Alison Walker, Violeta Golea, and Cindy Pearl for their revisions of the

code and ancillary data products and the testing of the revisions from D to H.

Edited by: David Carlson
Reviewed by: two anonymous referees

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

## Remarks from the language copy-editor

## Remarks from the typesetter

**TS20**  Please provide a statement on how your underlying research data can be accessed. If the data are not publicly accessible, a detailed explanation of why this is the case is required. The best way to provide access to data is by depositing them (as well as related metadata) in reliable public data repositories, assigning digital object identifiers (DOIs), and properly citing data sets as individual contributions. Please indicate if different data sets are deposited in different repositories or if data from a third party were used. If no DOI is available, assets can be linked through persistent URLs to the data set itself (not to the repositories' home page). This is not seen as best practice and the persistence of the URL must be secured.