# Peer review of "The International Satellite Cloud Climatology Project H-Series climate data record product"

_Earth System Science Data, 2017_

## Referee Comment (RC1) · Anonymous Referee #1 · 14 Sep 2017

1. Abstract: it would be useful (for users) to learn about main differences between D&H series (resolution, data format, impacts of algorithms . . .)

2. Table 1 shows the comparisons of ancillary products used by ISCCP for D- and H-series. It would be very useful to have one table showing the differences between D- & H-series ISCCP products. It is the most important thing from a user perspective.

3. L212-213: Pls. provide the reference for "ISCCP Climate-Algorithm Theoretical Basis Document".

4. The authors did a good job on listing the differences between D- & H-series in Sections 2-4, but it is lack of discussions on specific impacts of the changes for each

change. What should the users be aware of? Should they be concern about the validity of their conclusions made in their prior publications?

5. L233-238: What you said about "higher" or "lower" is opposite of what shows in Fig. 4 since Fig. 4 black line shows H-D. Pls. check.

6. Fig. 4: it seems that there is a shift in H-D differences around 1994. What causes it? Pls. add x-axis tick marks for all panels and add some vertical grid lines to make it easier to read years. Are there differences in estimated CF trends?

---

## Referee Comment (RC2) · Anonymous Referee #2 · 8 Nov 2017

Review results for the manuscript entitled "The International Satellite Cloud Climatology Project H - Series Climate Data Record Product" by Young et al.

I suggest rejecting the manuscript in its current form since a number of key elements are missing in the manuscript, e.g.:

1) The introduction should have a broad scope, summarizing other international activities (e.g. Patmos-X, CMSAF CLARA-A1/2, MODIS Collection 6, ESA Cloud_cci) and putting them in contrast to the characteristics of the dataset presented. This is completely missing.

2) The manuscript needs to describe also all other product variables (not only cloud

amount) more comprehensively, how are they retrieved, how do they look like (Showing examples of most of them), which caveats exist for them...

3) Validation. Datasets without any information about the accuracy of the products are useless to the users. High quality references observations exist, at least for some of the products variables: SYNOP: cloud amount, CALIOP: cloud mask/amount, cloud phase, cloud top pressure/height, CALIOP+CloudSat (e.g. DARDAR): IWP, AMSR-E: LWP. Basic validation results are mandatory for presenting a dataset.

---

## Editor Comment (EC1) · D. J. Carlson (Editor) · 21 Nov 2017

I insert this editorial comment to recognise and help resolve disparate recommendations of two reviewers, to add some review comments of my own, and to clarify the intent of ESSD.

First, I thank both reviewers for good efforts! Reviewing a data set for ESSD requires more time and more effort than reviewing a paper for a research journal. If, in this case, the review process took longer than expected, in the end we have two thoughtful reviews to evaluate.

[Figure]

As the open discussion forum shows, one review asks for clarification of the differences (improvements?) of the newly-submitted version of ISCCP data, e.g. H (new) vs D (prior), and explicit guidance for users adopting the newer product but recommends publication. A second review notes that the manuscript fails to explicitly evaluate the impact of ISCCP products in relation to other satellite-based cloud climatologies and lacks a validation section; this second reviewer recommends rejection.

I hope I understand both viewpoints but I also hope I see a larger picture. I see, for example, that the ESA Cloud CCI (the only serious 'competitor' to ISCCP from the list recited by reviewer 2, as the other products in that list represent narrower efforts focused on fewer sensors, generally - as the NCAR mirror of the EUMETSAT CM SAF product reports - "still not of sufficient quality to allow global climate trend analysis") has submitted its own data description to ESSD: ESSD-2017-48, accepted for publication. Some reasons for the disparities evident in the reviews emerge from comparison of these two data descriptions. We find one long-standing (ISCCP) and one very new (ESA Cloud CCI) product. ISCCP has decades of evaluation and scrutiny (including validation and invalidation) in the cloud and climate literature. ESA Cloud CCI seeks to establish their own identity and credentials with this first ESSD product. ISCCP quotes WCRP and CDR (climate data records) while ESA Cloud CCI quotes GCOS and ECV (essential climate variables). WCRP and GCOS represent close partners but, in addition to technical distinctions, CDR and ECV tend to have geographic identities: CDR prevalent in North America and US science agencies, ECV prominent in Europe and European science agencies. ISCCP present 3-hourly data at 10 km, but only at two wavelengths. ESA Cloud CCI presents multi-spectral data but mostly in monthly averages. ISCCP uses polar orbiting and geostationary satellites; ESA Cloud CCI focuses entirely on polar orbiters. ISCCP uses international satellite data streams to cover the globe, ESA Cloud CCI uses European and US satellites to cover 60N to 60S. Itemising these differences and their plausible influence on reviewer's viewpoints obscures a fundamental and very positive fact: through ESSD, users can get free and well-documented access to both data products!

[Figure]

(And, I note, to a third related data product on satellite-derived water vapour measurements - ESSD-2017-128, newly submitted - involving authors from both the ISCCP and ESA Cloud CCI communities.)

Review comments:

Take much more care with language. We read (page 3, line 51) that the D-Series product "has not been updated beyond December of 2009". But, the new H-Series data as presented also do not, as of this submission, extend beyond 2009. The sentence in question should read 'has not been updated since' Dec 2009? Or some other confusion intervenes? In the later section (Section 4.1, page 6) each description of an H-Series product includes some reference to the prior D-Series product. In some cases the authors write 'HXX represents the analog to DX'. That we can understand. In many other cases, however, we read that HXX "is like" DX. But, because of improved spatial resolution and other factors, HXX is different to, better than, but NOT like, DX? Choose a precise terminology and apply it in all comparison statements?

Recognise other contributions. The reader gets a strong sense of attention to ISCCP and recognition of ISCCP historical versions, impact and contributions, but no sense that this author team recognises any other similar or related efforts elsewhere? At least give the reader some sense that you pay attention to other efforts? You should cite ESSD-2017-48 as it predates your contribution in the same journal? Use ESSD-2017-48 as an example? What do they include that you might also include?

Show ISCCP as a fresh relevant contribution. Google Analytics showing that a 25 year-old paper, Rossow and Schiffer 1991, has 1500 citations, does not send the message you want. If you want to use citation statistics, perhaps citations of ISCCP since 2015 from Google Scholar? I did not find Figure 1 useful or informative.

In an additional paragraph or two, answer both the 'what do I need to know to use this new version' question from reviewer 1 and, by citing key papers from the ISCCP literature, at least show that you know that other researchers have evaluated and reported
on accuracy and validation questions from reviewer 2.

In summary, for ESSD, I tend to think the ISCCP submission qualifies as a valid and useful data product. For ESSD we expect authors to provide an interesting data set with potential wide application, with sufficient detail to assure quality and reproducibility, and with sufficient graphic or tabular examples to demonstrate quality and utility. We recognise and support the probability of periodic data updates. ESSD does not invite or expect full rigorous scientific analysis of the data. Taking the present examples, ESSD encourages ESA Cloud CCI and ISCCP to describe and share their data but we would not expect either to conduct a full intercomparison with the other. That intercomparison very likely represents an important and valuable scientific contribution, but for publication in ACP, Jnl of Climate or BAMS, not ESSD. If the ESSD process has succeeded, researchers conducting that intercomparison will enjoy open access and detailed descriptions.

---

## Author Comment (AC1) · 13 Dec 2017

Response to Reviewer #1 (In Blue) 1. Abstract: it would be useful (for users) to learn about main differences between D&H series (resolution, data format, impacts of algorithms . . .) Response_1: Please see lines 31-35 of the abstract. They provide the following language. "Key refinements included in the ISCCP H-Series CDR are based on improved quality control measures, modified ancillary inputs, higher spatial resolution input and output products, calibration refinements, and updated documentation and metadata to bring the H-Series product into compliance with existing standards for climate data records." This info is a general description of the

changes from the D-Series to the H-Series product. 2. Table 1 shows the comparisons of ancillary products used by ISCCP for D- and H-series. It would be very useful to have one table showing the differences between D- & H-series ISCCP products. It is the most important thing from a user perspective. Response_2: As the reviewer mentions (later) in Comment #4. The paper does identify the differences between the D-&H-Series product. However, an additional table (Table 2) has been added to highlight the differences between the H and D series products in the updated draft of the paper. 3. L212-213: Pls. provide the reference for "ISCCP Climate-Algorithm Theoretical Basis Document". Response_3: The reference for the ISCCP C-ATBD has been added. 4. The authors did a good job on listing the differences between D- & H-series in Sections 2-4, but it is lack of discussions on specific impacts of the changes for each. What should the users be aware of? Should they be concern about the validity of their conclusions made in their prior publications? Response_4: This paper introduces the users to the new H-Series product. We make very few updates to correct issues that users have previously described in other papers. There are plans to address some calibration related issues in future re-processing. However, at this time, the v01r00 of the CDR product is basically the same D-product but processed at the higher spatial resolution. Any known issues for the D-Series product would then apply to the H-Series product. This point is referenced in Lines 255-256. 5. L233-238: What you said about "higher" or "lower" is opposite of what shows in Fig. 4 since Fig. 4 black line shows H-D. Pls. check. " Response_5: The black line in Fig 4a-c shows the differences for H-D. For the global case the average monthly differences are 0.21%. This means that H-Series Global Cloud Fraction is greater (higher) than D-Series. Please let me know if I misunderstand your point. 6. Fig. 4: it seems that there is a shift in H-D differences around 1994. What causes it? Response_6: This appears to be an optical effect caused by the difference line crossing the CF anomaly lines – if you look carefully there is no real change in CF anomaly between D and H versions. 7. Pls. add x-axis tick marks for all panels and add some vertical grid lines to make it easier to read years. Response_7: These updates have been completed. Are there

differences in estimated CF trends? This is a dataset paper, we simply introduce the new H-Series and provide similarities with the D-Series product. We do not argue for or against trends in CF or other cloud properties. However, we do list references that have evaluated CF trends in the ISCCP D-Series products. Trends should be evaluated with caution. 8. Are there differences in estimated CF trends? Response_8: This is a dataset paper, we simply introduce the new H-Series and provide similarities with the D-Series product. We do not argue for or against trends in CF or other cloud properties. However, we do list references that have evaluated CF trends in the ISCCP D-Series products. Trends should be evaluated with caution.

Please also note the supplement to this comment:
https://www.earth-syst-sci-data-discuss.net/essd-2017-73/essd-2017-73-AC1-
supplement.pdf

**Supplement:**

Response to Reviewer #1 (In Blue)

1. Abstract: it would be useful (for users) to learn about main differences between D&H series (resolution, data format, impacts of algorithms . . .) Please see lines 31-35 of the abstract. They provide the following language. "Key refinements included in the ISCCP H-Series CDR are based on improved quality control measures, modified ancillary inputs, higher spatial resolution input and output products, calibration refinements, and updated documentation and metadata to bring the H-Series product into compliance with existing standards for climate data records." This info is a general description of the changes from the D-Series to the H-Series product.

2. Table 1 shows the comparisons of ancillary products used by ISCCP for D- and H-series. It would be very useful to have one table showing the differences between D- & H-series ISCCP products. It is the most important thing from a user perspective. As the reviewer mentions in Comment #4, the paper does identify the differences between the D-&H-Series product. An additional table highlighting the differences between the H and D series products and the impacts of these changes has been provided in an updated draft of the paper and are now included in Table 2.

3. L212-213: Pls. provide the reference for "ISCCP Climate-Algorithm Theoretical Basis Document". The reference for the ISCCP C-ATBD has been added.

4. The authors did a good job on listing the differences between D- & H-series in Sections 2-4, but it is lack of discussions on specific impacts of the changes for each. What should the users be aware of? Should they be concern about the validity of their conclusions made in their prior publications? The dataset paper introduces the users to the new H-Series product. We do not make any updates to correct issues that users have previously described in other papers. There are plans to address some calibration related issues in future re-processing. However, at this time, the v01r00 of the CDR product is basically the same D-product but processed at the higher spatial resolution. Any known issues for the D-Series product would then apply to the H-Series product. This point is referenced in Lines 295-301.

5. L233-238: What you said about "higher" or "lower" is opposite of what shows in Fig. 4 since Fig. 4 black line shows H-D. Pls. check. " The black line in Fig 4a-c shows the time series of differences for H-D. For the global case the average monthly differences are 0.21%. This means that H-Series Global Cloud Fraction is greater (higher) than D-Series. Please let me know if I misunderstand your point.

6. Fig. 4: it seems that there is a shift in H-D differences around 1994. What causes it? This appears to be an optical effect caused by the difference line crossing the CF anomaly lines -- if you look carefully there is no real change in CF anomaly between D and H versions.

7. Pls. add x-axis tick marks for all panels and add some vertical grid lines to make it easier to read years. These updates have been completed.

8. Are there differences in estimated CF trends? This is a dataset paper, we simply introduce the new H-Series and provide similarities with the D-Series product. We do not argue for or against trends in CF or other cloud properties. However, we do list references that have evaluated CF trends in the ISCCP D-Series products. Trends should be evaluated with caution.

|  | D-Series | H-Series | Impacts |
|---|---|---|---|
| **Input Resolution** | 30-km, B3 | 10-km, B1U | Higher spatial resolution output products |
| **Cloud Algorithm** | Combination of IR and VIS, and NIR channels in polar regions | Only IR and VIS for entire period of record | Reduces low-level cloud sensitivity over snow and ice in polar regions |
| **Metadata** | N/A | CF 1.6 | Improved documentation for product reproducibility |
| **Improved/Added Ancillary Data** | | Example: Macv1 Aerosols Product | Increases cloud fraction due to volcanic eruptions and reduce cloud optical thicknesses in areas with larger aerosol abundances |
| **Period of Record** | 07/1983–12/2009 w/ no additional production planned to extend POR | 07/1983–12/2009 w/ additional production planned to extend POR | Restores research application of ISCCP data post 2009 |
| **Format** | Binary | Netcdf for all products except HXS | Supported by netcdf software applications and tools |
| **Available Products** | DX (~30 km, 3hrly) | HXS (~10 km, 3hrly) | |
| | N/A | HXG (0.1°, 3hrly) | |
| | DS | HGS (1.0°, 3hrly) | |
| | D1 (2.5°, 3hrly) | HGG (1.0°, 3hrly) | |
| | D2 (2.5°, 3-hrly monthly avg.) | HGH (1.0°, 3hrly monthly avg.) | |
| | D3 (2.5°, monthly avg.) | HGM (1.0°, monthly avg.) | |

Table 2: High-level summary of differences between ISCCP D-Series and H-Series products and their impacts. Other details on differences are provided

---

## Author Comment (AC2) · 13 Dec 2017

Re: Interactive (EC1) comment on "The International Satellite Cloud Climatology Project H-Series Climate Data Record Product" by Alisa H. Young et al.

Thank you for your comments.

EC1. Take much more care with language. We read (page 3, line 51) that the D-Series product "has not been updated beyond December of 2009". But, the new H-Series data as presented also do not, as of this submission, extend beyond 2009. The sentence in question should read 'has not been updated since' Dec 2009? Or some other confusion

intervenes?

Response_1:On page 3 Line 51 of the updated manuscript, the language has been revised to show that the product has not been updated since Dec. of 2009.

EC2. In the later section (Section 4.1, page 6) each description of an H-Series product includes some reference to the prior D-Series product. In some cases the authors write 'HXX represents the analog to DX'. That we can understand. In many other cases, however, we read that HXX "is like" DX. But, because of improved spatial resolution and other factors, HXX is different to, better than, but NOT like, DX? Choose a precise terminology and apply it in all comparison statements?

Response_2: The language in Section 4.1 has been revised to better illustrate the parallels between the D-Series and H-Series products. The language you've mentioned that could cause confusion, has been removed/replaced with more appropriate terminology. In addition, other parts of the text have been updated to take better care in describing the ISCCP H-Series product, processing, and updates.

EC3: Recognize other contributions. The reader gets a strong sense of attention to ISCCP and recognition of ISCCP historical versions, impact and contributions, but no sense that this author team recognises any other similar or related efforts elsewhere? At least give the reader some sense that you pay attention to other efforts? Sections 1 and 2 of the paper now have more references to other cloud datasets and work that has been done to evaluate ISCCP. The language now provides more context regarding cloud datasets, and where the ISCCP products fit within that general schema.

Response3: Please note the following text that was written on this issue in response to Reviewer #2 "…..However, the updated manuscript has been modified to address the Reviewer's concerns. The Introduction of the paper now contains more references to other cloud datasets and work that has been done to evaluate global cloud characteristics and ISCCP. The language now provides more context regarding a broader scope of other cloud datasets, and addresses where the ISCCP products fit within that

general schema. "

The following references have been added: Cao, C., De Luccia, F. J., Xiong, X., Wolfe, R., and Weng, F.: Early on-orbit performance of the visible infrared imaging radiometer suite onboard the Suomi National Polar-Orbiting Partnership (S-NPP) satellite,

Evan, A. T., Heidinger, A. K., and Vimont, D. J.: Arguments against a physical long‐term trend in global ISCCP cloud amounts, Geophysical Research Letters, 34(4), 2007. Hutchison, K. D., Roskovensky, J. K., Jackson, J. M., Heidinger, A. K., Kopp, T. J., Pavolonis, M. J., and Frey, R.: Automated cloud detection and classification of data collected by the Visible Infrared Imager Radiometer Suite (VIIRS), International Journal of Remote Sensing, 26(21), 4681-4706, 2005.

Jiménez, C., Prigent, C., Catherinot, J., Rossow, W., Liang, P. and Moncet, J.L.: A comparison of ISCCP land surface temperature with other satellite and in situ observations, Journal of Geophysical Research-Atmospheres, 117(D8), 2012.

Norris, J. R.: What can cloud observations tell us about climate variability? Space Sci. Rev., 94(1–2), 375–380, 2000.

Platnick, S., King, M. D., Ackerman, S. A., Menzel, W. P., Baum, B. A., Riédi, J. C., and Frey, R. A.: The MODIS cloud products: Algorithms and examples from Terra, IEEE Transactions on Geoscience and Remote Sensing, 41(2), 459-473, 2003.

Raschke, E., Bakan, S., and Kinne, S.: An assessment of radiation budget data provided by the ISCCP and GEWEX‐SRB, Geophysical Research Letters, 33(7), 2006.

Stengel, M., Stapelberg, S., Sus, O., Schlundt, C., Poulsen, C., Thomas, G., Christensen, M., Henken, C.C., Preusker, R., Fischer, J. and Devasthale, A.: Cloud property datasets retrieved from AVHRR, MODIS, AATSR and MERIS in the framework of the Cloud_cci project, Earth System Science Data, 9(2), 881, 2017.

Stubenrauch, C.J., Rossow, W.B., Kinne, S., Ackerman, S., Cesana, G., Chepfer, H., Di Girolamo, L., Getzewich, B., Guignard, A., Heidinger, A. and Maddux, B.C.: Assessment of global cloud datasets from satellites, A Project of the World Climate Research Programme Global Energy and Water Cycle Experiment (GEWEX) Radiation Panel, 2012.

Stubenrauch, C.J., Rossow, W.B., Kinne, S., Ackerman, S., Cesana, G., Chepfer, H., Di Girolamo, L., Getzewich, B., Guignard, A., Heidinger, A. and Maddux, B.C.: Assessment of global cloud datasets from satellites: Project and database initiated by the GEWEX radiation panel, Bulletin of the American Meteorological Society, 94(7), 1031-1049, 2013.

EC4: You should cite ESSD-2017-48 as it predates your contribution in the same journal? Response_4: We now cite ESSD-2017-48 in the introduction of the text where more references are now included to highlight a broader collection of other cloud datasets. (Please see added references listed above).

EC5: Use ESSD-2017-48 as an example? What do they include that you might also include? Response_5: ESSD-2017-48 is a good paper. However, it introduces a new dataset. ISCCP is not new although the H-Series product does provide a new version of the dataset with updates outlined in the text. There is much documentation on ISCCP. Thus, the specific elements captured in the text for ESSD-2017-48 does not need to be captured to the same degree for the ISCCP H-Series paper. Moreover, the paper is designed to highlight the general updates of the product. The reader may also refer to the full 179pp ISCCP H-Series Climate-Algorithm Theoretical Basis Document (C-ATBD) which is already publicly available.

EC6: Show ISCCP as a fresh relevant contribution. Google Analytics showing that a 25 year old paper, Rossow and Schiffer 1991, has 1500 citations, does not send the message you want. Response_6: The message that we would like to send is that ISCCP has a long legacy. Although, the figure is not new and informative for this particular reader, you assume that all users will be aware of its history and relevance. However, this is not the case. Figure 1, provides some context regarding why continu-

ing ISCCP, beyond its 2009 product updates is beneficial to the climate and modeling communities and also highlights its accomplishments as a dataset. Yes, the earlier references are old. However, all papers do not have this frequency of reference. Do citations, no longer mean anything? I argue that they do and that it is highly valuable and worth continuation, despite the (fixable) flaws within the product and its competition with other datasets that rely on more advanced methods based on technologically superior instruments.

EC7: If you want to use citation statistics, perhaps citations of ISCCP since 2015 from Google Scholar? I did not find Figure 1 useful or informative. Response_7: Please see previous response.

EC8:In an additional paragraph or two, answer both the 'what do I need to know to use this new version' question from reviewer 1 and, by citing key papers from the ISCCP literature, at least show that you know that other researchers have evaluated and reported on accuracy and validation questions from reviewer 2. Response_8: These points have been addressed. Table 2 has been added to satisfy Reviewer 1 who requested that the key differences between the D and H-Series products be better highlighted and again, additional language has been added in the Introduction to highlight the contributions of other datasets for cloud detection and retrieval.

Please also note the supplement to this comment:
https://www.earth-syst-sci-data-discuss.net/essd-2017-73/essd-2017-73-AC2-supplement.pdf
* * *
]

---

## Author Comment (AC3) · 13 Dec 2017

RE: Interactive comment on "The International Satellite Cloud Climatology Project H-Series Climate Data Record Product" by Alisa H. Young et al.  Anonymous Referee #2

Review results for the manuscript entitled "The International Satellite Cloud Climatology Project H - Series Climate Data Record Product" by Young et al.  I suggest rejecting the manuscript in its current form since a number of key elements are missing in the

manuscript, e.g.:

RC1: 1) The introduction should have a broad scope, summarizing other international activities (e.g. Patmos-X, CMSAF CLARA-A1/2, MODIS Collection 6, ESA Cloud_cci) and putting them in contrast to the characteristics of the dataset presented. This is completely missing.

Response_1: None of the listed "activities" is international and none have a time resolution and record length comparable to ISCCP. In any case, most of these other products were compared to the D-version ISCCP product by the GEWEX Assessment, which is summarized by Stubenrauch et al 2013 and described in more detail in a larger report available at http://www.wcrp-climate.org/documents/GEWEX_Cloud_Assessment_2012.pdf. So, to respond to the concerns addressed in 1) the text has been modified to mention the GEWEX Assessment without a detailed listing of all other projects, which seems unnecessary. However, the updated manuscript has been modified to address the Reviewer's concerns. The Introduction of the paper now contains more references to other cloud datasets and work that has been done to evaluate global cloud characteristics and ISCCP. The language now provides more context regarding a broader scope of other cloud datasets, and addresses where the ISCCP products fit within that general schema.

The following references have been added: Cao, C., De Luccia, F. J., Xiong, X., Wolfe, R., and Weng, F.: Early on-orbit performance of the visible infrared imaging radiometer suite onboard the Suomi National Polar-Orbiting Partnership (S-NPP) satellite,

Evan, A. T., Heidinger, A. K., and Vimont, D. J.: Arguments against a physical long‐term trend in global ISCCP cloud amounts, Geophysical Research Letters, 34(4), 2007. Hutchison, K. D., Roskovensky, J. K., Jackson, J. M., Heidinger, A. K., Kopp, T. J., Pavolonis, M. J., and Frey, R.: Automated cloud detection and classification of data collected by the Visible Infrared Imager Radiometer Suite (VIIRS), International Journal of Remote Sensing, 26(21), 4681-4706, 2005.

Jiménez, C., Prigent, C., Catherinot, J., Rossow, W., Liang, P. and Moncet, J.L.: A comparison of ISCCP land surface temperature with other satellite and in situ observations, Journal of Geophysical Research-Atmospheres, 117(D8), 2012.

Norris, J. R.: What can cloud observations tell us about climate variability? Space Sci. Rev., 94(1–2), 375–380, 2000.

Platnick, S., King, M. D., Ackerman, S. A., Menzel, W. P., Baum, B. A., Riédi, J. C., and Frey, R. A.: The MODIS cloud products: Algorithms and examples from Terra, IEEE Transactions on Geoscience and Remote Sensing, 41(2), 459-473, 2003.

Raschke, E., Bakan, S., and Kinne, S.: An assessment of radiation budget data provided by the ISCCP and GEWEX‐SRB, Geophysical Research Letters, 33(7), 2006.

Stengel, M., Stapelberg, S., Sus, O., Schlundt, C., Poulsen, C., Thomas, G., Christensen, M., Henken, C.C., Preusker, R., Fischer, J. and Devasthale, A.: Cloud property datasets retrieved from AVHRR, MODIS, AATSR and MERIS in the framework of the Cloud_cci project, Earth System Science Data, 9(2), 881, 2017.

Stubenrauch, C.J., Rossow, W.B., Kinne, S., Ackerman, S., Cesana, G., Chepfer, H., Di Girolamo, L., Getzewich, B., Guignard, A., Heidinger, A. and Maddux, B.C.: Assessment of global cloud datasets from satellites, A Project of the World Climate Research Programme Global Energy and Water Cycle Experiment (GEWEX) Radiation Panel, 2012.

Stubenrauch, C.J., Rossow, W.B., Kinne, S., Ackerman, S., Cesana, G., Chepfer, H., Di Girolamo, L., Getzewich, B., Guignard, A., Heidinger, A. and Maddux, B.C.: Assessment of global cloud datasets from satellites: Project and database initiated by the GEWEX radiation panel, Bulletin of the American Meteorological Society, 94(7), 1031-1049, 2013.

EC2: 2) The manuscript needs to describe also all other product variables (not only cloud amount) more comprehensively, how are they retrieved, how do they look like

(Showing examples of most of them), which caveats exist for them...

Response_2: The ISCCP Product has many variables. However, the cloud variables listed in the description of the text are the most widely used. A more comprehensive description of these variables are documented in other places including the C-ATBD. Thus the manuscript focuses on the updates to the ISCCP algorithm. A general description of the product's caveats are given.

Please also note the supplement to this comment:
https://www.earth-syst-sci-data-discuss.net/essd-2017-73/essd-2017-73-AC3-supplement.pdf